# Exploring factors impacting Hispanic/Latinx individuals' response to a type 2 diabetes digital storytelling intervention

Abby M. Lohr[1,2]*, Sunny W. Kim[3], Jennifer St. Sauver[2], Jamie O'Byrne[2],
Joan M. Griffin[4], Rozalina G. McCoy[5,6], John M. Ruiz[7], Irene G. Sia[8], Mark L. Wieland[1,9]

1 Center for Clinical and Translational Science, Mayo Clinic, Rochester, Minnesota, United States of America, 2 Department of Quantitative Health Sciences, Division of Epidemiology, Mayo Clinic, Rochester, Minnesota, United States of America, 3 Edson College of Nursing and Health Innovation, Arizona State University, Phoenix, Arizona, United States of America, 4 Robert D. and Patricia E. Kern Center for the Science of Health Care Delivery, Mayo Clinic, Rochester, Minnesota United States of America, 5 University of Maryland Institute for Health Computing, Bethesda, Maryland, United States of America, 6 Division of Endocrinology, Diabetes & Nutrition, Department of Medicine, University of Maryland School of Medicine, Baltimore, Maryland, United States of America, 7 Department of Psychology, College of Science, University of Arizona, Tucson, Arizona, United States of America, 8 Division of Public Health, Infectious Diseases, and Occupational Medicine, Mayo Clinic, Rochester, Minnesota United States of America, 9 Division of Community Internal Medicine, Geriatrics, and Palliative Care, Department of Medicine, Mayo Clinic, Rochester, Minnesota, United States of America

* lohr.abby@mayo.edu

## Abstract

### Background

Hispanic/Latinx individuals have high prevalence of type 2 diabetes and its complications yet often face barriers in accessing diabetes prevention and self-management interventions. One possible approach is to implement digital storytelling interventions, which involve narrative-driven videos made by individuals who have lived experience with particular conditions or illnesses. These stories can inspire viewers with similar life experiences to change behaviors or attitudes. Little is known about which characteristics influence how individuals respond to digital storytelling interventions with healthful behaviors and improved outcomes – information necessary to further tailor these interventions to improve type 2 diabetes outcomes. Previously, the Rochester Healthy Community Partnership used the digital storytelling process to develop Stories for Change Diabetes and tested intervention effectiveness.

### Methods

We conducted a secondary analysis to examine the sociodemographic and disease-related factors that affected participants' responses to the Stories for Change Diabetes intervention. Drawing on Social Cognitive Theory and Culture-Centric Health Promotion principles, we analyzed results from the 227 intervention

**Data availability statement:** Data are available in a Zenodo repository available here: https://doi.org/10.5281/zenodo.14618704

**Funding:** Research reported in this manuscript was supported by the National Institute of Diabetes and Digestive and Kidney Diseases (https://www.niddk.nih.gov/) of the National Institutes of Health under award number F32DK135200. The funders had no role in study design, data collection and analysis, decision to publish, or preparation of the manuscript.

**Competing interests:** The authors declare that no competing interests exist.

participants stratified by whether they experienced a clinically meaningful decrease (>0.5%) in hemoglobin A1c between baseline and three-month follow-up. We then used multivariable logistic regressions to identify factors associated with change in hemoglobin A1c.

## Results

Participants with diabetes duration <5 years and/or whose diabetes self-efficacy improved between baseline and 3-month follow-up were more likely to experience a meaningful decrease in hemoglobin A1c at three months (compared to participants without those characteristics).

## Conclusions

These findings will provide insight into how digital storytelling interventions can be effectively tailored to Hispanic/Latinx individuals most likely to benefit.
Trial registration: Not applicable

## Introducvtion

In 2021, the diabetes mortality rate in the United States (US) was 29 deaths per 100,000 among Hispanic/Latinx individuals, while it was 22 for Non-Hispanic Whites [1]. This preventable loss of life is both heartbreaking and expensive for families and healthcare systems. Given the disproportionate burden of type 2 diabetes (T2D) on the Hispanic/Latinx community, urgent action is warranted to prevent diabetes and its complications and improve management. This is especially crucial given that the Hispanic/Latinx population is the largest and the second fastest growing ethnic minority group in the US [2].

Reducing T2D-related disparities among Hispanic/Latinx populations will require tailored interventions. One possible approach is using digital storytelling (DST). Storytelling is a specific cultural narrative showing promise as an effective strategy for promoting health behavioral changes [3]. Digital stories, as produced in DST workshops, are narrative-driven videos created by individuals who share their firsthand experiences. Storytellers co-create their narratives by writing scripts, selecting images and sounds, participating in video editing, and sharing their digital stories with the group. The end result consists of an audio-visual narrative, accompanied by still images and voice-over narration, lasting 2–5 minutes [4]. DST differs from other narrative interventions by placing the individuals affected by the theme at the heart of the knowledge creation process. DST workshops foster a sense of community with others who have similar experiences, which may enrich the cultural relevance of the stories created. Moreover, digital stories can provide an alternative perspective to daily discourse by elevating the voices of historically marginalized individuals, opening the tools and space to represent themselves.

To elaborate on the DST process, with storytellers' consent researchers can employ digital stories as a health promotion intervention for individuals who were

not involved in the story creation [5,6]. These culturally resonant stories can be a catalyst for change by inspiring not only the storytellers themselves but also viewers with similar life experience (heretofore called 'DST interventions'). Through DST interventions, these viewers may engage with characters, stories, and cultural elements, leading to change in health behaviors or attitudes [3,7,8].

While there is some research on the impact of participating in the DST process [9–11] and the effects of narrator point of view [12,13], little is known about which sociodemographic and disease characteristics influence how viewers respond to DST interventions. There is a need to investigate how the effect of DST interventions on viewers' reactions vary to further tailor these interventions to Hispanic/Latinx individuals and determine what other complementary interventions are needed to achieve T2D desired outcomes.

## Parent study

Since 2004, Rochester Healthy Community Partnership (RHCP) has used a community-based participatory research (CBPR) approach alongside immigrant and refugee communities in southeast Minnesota [14]. In 2014, Hispanic/Latinx community partners expressed that the systems for T2D management did not address the sociocultural or linguistic needs of their communities which negatively impacted health outcomes. Community partners also implicated low health literacy as a common pathway for sub-optimal T2D care. To address these concerns, RHCP used the DST process to develop the Stories for Change (S4C) Diabetes intervention [15–17]. RHCP recruited four Hispanic/Latinx individuals diagnosed with T2D to share their stories. Storytellers completed a story building workshop with RHCP staff and two facilitators from StoryCenter [18]. At the end of the workshop, participants created culturally relevant stories in Spanish about their personal experiences related to physical activity, dietary quality, medication adherence, and blood glucose self-monitoring. Storytellers provided consent for the dissemination of the final products. Given the culturally tailored nature of S4C, this intervention is specifically designed for Hispanic/Latinx adults diagnosed with T2D.

RHCP piloted the digital stories as an intervention among 13 Hispanic/Latinx adults diagnosed with T2D. Pilot results revealed that participants in the intervention group had a hemoglobin A1C (HbA1c) that was 37% times the standard deviation lower than those in the control group, indicating a small to medium effect size [16]. This encouraging preliminary data prompted the creation and funding of a randomized clinical trial to assess the effectiveness of the S4C intervention (R01DK113999) [17], which showed a modest improvement in glycemic control after three months (clinicaltrials.gov ID NCT03766438).

## Theoretical approach

The development of the S4C intervention was guided by blending two theoretical perspectives: the Social Cognitive Theory (SCT) and storytelling as Culture-Centric Health Promotion Model (CCHP) [19]. CCHP applied to health promotion in the context of cultural alignment describes the process of developing behavioral change mechanisms through stories [20]. In this model, we suggest that culturally resonant stories show desired health behaviors and transport participants through emotional engagement. They are "carried away," and identify with the characters, story, or cultural elements. When individuals feel emotionally connected to and identify with characters, role modeling in narratives can result in attitudinal and/or behavior change [19].

Transportation and identification are posited to heighten elements of SCT constructs including (1) observational learning or a person's ability to witness and successfully repeat behaviors; and (2) self-efficacy or a person's confidence in their ability to successfully perform behaviors [21]. Observational learning and self-efficacy are associated with T2D self-care behaviors [22–24].

Our approach of integrating SCT and CCHP is appropriate because these theoretical constructs complement each other. We predict that S4C participants would be emotionally engaged by a narrative about T2D self-management because they identify with the Hispanic/Latinx characters in the digital stories. Through observational learning (SCT) and

transcription (CCHP), participants can gain insights, skills, and inspiration. This enhanced understanding and engagement, in turn, can improve their self-efficacy (SCT) and facilitate behavioral change, leading to improved T2D self-management skills and ultimately improved disease management and outcomes.

## Present study

In this investigation, we use the S4C randomized control trial data to conduct a secondary analysis of the participants who received the intervention (n = 227) to identify sociodemographic and disease status characteristics that influenced how Hispanic/Latinx individuals with T2D responded to S4C. Control group participants, who did not view the intervention, were excluded from this analysis. We hypothesized that the S4C intervention would have a greater impact on individuals with lower levels of formal education and higher baseline HbA1c. This hypothesis was based on previous research from other diseases showing that people with this profile benefit more from DST interventions compared to those with more formal education [25] and more optimal disease metrics [16,26]. Beyond education and disease status, no other associations have been found between sociodemographic characteristics or disease metrics and viewers' response to digital stories [27].

## Materials and methods

### Design overview

The S4C project was a two-group, parallel randomized controlled trial conducted in primary care settings across two healthcare institutions. The study included 450 Hispanic/Latinx adults with elevated HbA1c (≥8%) [15]. Participants in the intervention group watched the 12-minute digital story in addition to receiving usual clinical care while the comparison group received only usual clinical care (visits with their primary care provider, diabetes educator, pharmacist, or promotra as needed). The two clinical sites were Hennepin Healthcare in Minneapolis, Minnesota and Mountain Park Health Center in Phoenix, Arizona. S4C completed enrollment in July 2022. Both the parent and current studies were approved by the Mayo Clinic Institutional Review Board (18–002998 and 22–008310).

### Participants

Eligibility criteria were 1) self-identify as Hispanic/Latinx, 2) age 18–70 years, 3) received primary care at a participating clinical site, 4) at least one office visit with their primary care team within the las year, 5) T2D diagnosis, 6) T2D duration ≥6 months, 7) most recent HbA1c ≥ 8% (within one month of enrollment), and 8) intention to continue receiving medical care at the recruitment clinic for the next six months (which was established via survey question).

Study staff used institutional diabetes registries at the participating sites to identify eligible participants and recorded their upcoming diabetes-related appointments. They then reached out to eligible individuals to assess their interest in study participation.

### Study procedures

The baseline study visit was conducted in the participants' preferred language (English or Spanish). During this visit, staff confirmed participants' eligibility, obtained informed consent, collected biometric measurements, and administered surveys to obtain secondary outcomes and theory-based measures. Staff administered the same survey again at six weeks (via telephone) and three months (study visit) and repeated biometric data collection at three months.

### Measures

**Primary outcome.** The primary outcome was glycemic management as assessed by HbA1c measured from blood samples obtained and analyzed by the laboratories at the participating sites. A meaningful decrease in HbA1c was defined as a greater than 0.5% reduction from baseline to 3-month follow-up.

**Demographic variables.** We studied the following participant characteristics: sex (male/female), years living with diabetes (<5, 5–9, 10–19, 20+), education (high school or more, less than high school) and age (18–40, >40–55, >55–70 years).

**Additional biometric variables.** We collected additional biometric data including blood pressure, body mass index (BMI), and cholesterol. Study personnel recorded seated blood pressure on participants' right arms after they had rested quietly for five minutes. Blood pressure was measured three times, and the average of the second and third readings was used for analysis. Weight was measured to the nearest 0.1 kg using a portable scale, and height was measured to the nearest 0.1 cm at baseline using a stadiometer, with participants removing their shoes for both measurements. Body mass index (BMI) was calculated as weight (kg) divided by height squared (m²). Low-density lipoprotein (LDL) cholesterol was measured using the same blood sample that was utilized to determine HbA1c levels. LDL cholesterol was calculated for each participant based on these values: LDL cholesterol = total cholesterol – HDL cholesterol – (triglycerides/5). All measurements were collected for the study (and hence free to the patient) rather than for clinical care.

**Survey variables.** We evaluated the constructs of story identification and transportation from the Culture-Centric Health Promotion Model using the Narrative Quality Assessment Tool (NQAT). The tool consists of two 6-item subscales: one measuring identification with the story and the other assessing engagement (i.e., transportation) [20]. Story identification refers to the sense of connection with the storyteller, while transportation reflects the level of engagement with the narrative. NQAT responses range from disagree a lot [1] to agree a lot [5], with higher scores indicating greater identification and transportation. Each subscale is scored by calculating the mean score (score range: 1–5). This scale has good construct validity among Spanish-speaking populations [20].

We measured health literacy using the Short Assessment of Health Literacy – Spanish English [28]. The instrument includes 18 questions used to assess participants' ability to understand common medical terms. Participants are presented with three words and asked which two are related (e.g., kidney, urine, fever). The instrument is scored by totaling the number of correct responses, with each right answer earning one point (score range: 0–18). A score between 0 and 14 indicates that the participant may have low health literacy.

We assessed social support for diabetes using six items from Section V – Support Questions 2 (a-f) of the Diabetes Care Profile (DCP) [29]. Responses to these questions ranged from 1 (strongly disagree) to 5 (strongly agree). The scoring for this section involves summing the responses and dividing by the number of non-missing items. Higher scores reflect greater social support for diabetes-related activities.

We evaluated diabetes-related self-efficacy using a modified version of the Diabetes Self-Efficacy Scale (DSES) [30,31]. This 8-item questionnaire uses a Likert-type response scale ranging from 1 (not at all confident) to 10 (totally confident). The DSES is scored by averaging the responses across the eight items, with higher scores reflecting greater self-efficacy in managing diabetes (score range: 1–10).

It is permissible to use and modify the DSES. After completing our CBPR process of survey instrument translation [32], our community partners determined that questions 1 ("How confident do you feel that you can eat your meals every 4 to 5 hours every day, including breakfast every day?") and 5 ("How confident do you feel that you can do something to prevent your blood sugar level from dropping when you exercise?") were not a good fit and would be difficult to translate for our priority population. Following their recommendations, we excluded both questions from our questionnaire. Consequently, we conducted mean imputation: we assumed that, if participants had been asked questions 1 and 5, they would have had answers similar to their other responses. In cases like this (non-response to individual questions), Fairclough and Cella found that substituting the mean of the completed items is generally the most unbiased and precise approach when the completion rate is greater than 50% [33]. Our DSES completion rate was 100% (likely due to careful survey administration by study staff) and we therefore did not remove any participants from our analysis. Because we disrupted a known scale by removing two questions, we also calculated the Cronbach's Alpha at baseline (0.65) and 3-month follow-up (0.74) to

demonstrate reliability. Although the Cronbach's Alpha at baseline is lower than desired (≥ 0.70), this reduction in reliability is offset by the decreased burden from having participants answer fewer questions.

## Analysis

**Descriptive statistics.** In this secondary analysis, we analyzed results only from the 227 participants who received the intervention. We stratified participants into two groups: participants who did and did not experience a meaningful decrease (>0.5%) in HbA1c between baseline and the three-month follow-up. In a previous analysis [17], we examined the relationship between study site and change in HbA1c and did not find a significant association. Thus, in this analysis, we did not stratify or control for site. We examined the relationship between covariates and the primary outcome. We summarized categorial variables using counts and proportions. Continuous variables were checked for deviation from normality, and none deviated significantly therefore means and standard deviations were reported.

**Univariate and multivariate analyses.** Associations between predictor variables and meaningful change in HbA1c were assessed using Chi-square or Kruskall-Wallis tests, as appropriate. Next, univariate logistic regression models were used to measure associations between each predictor variable and meaningful change in HbA1c. Multivariate logistic regression was then performed adjusting for sex, age, years living with diabetes, education, biometric data, Social Cognitive Theory constructs, Culture-Centric Health Promotion Model constructs, and health literacy. Statistical analyses were conducted using SAS version 9.4 (SAS Institute; Cary, NC). All tests were two-sided, and p-values <0.05 were considered statistically significant.

## Results

### Study participants

Of the 227 participants randomized to the S4C intervention arm, about half (55%) experienced a meaningful decrease in HbA1c from baseline to 3 months. Shown in Table 1, more females compared to males participated in the study. Most participants who did not experience a meaningful decrease in HbA1c were between 55 and 70 years old (54%) while most of the participants who did experience a meaningful decrease were between 40 and 55 years (53%). In both groups, the plurality of participants had been living with diabetes for 10–19 years (39% among those who did not vs. 37% among those who did experience a meaningful decrease in HbA1c). Over one-quarter of each group completed formal schooling beyond high school. Participants in both groups had similar systolic and diastolic blood pressure, BMI, and LDL cholesterol at baseline.

In univariate analysis, participants who were >40 years old and had higher HbA1c levels at baseline were more likely to have a meaningful decline compared to younger participants or those with lower levels at baseline. In addition, participants with lower diabetes self-efficacy scores at baseline, but bigger changes in self-efficacy after completing the intervention were more likely to have meaningful declines.

### Associations between meaningful decrease in HbA1c and participant characteristics

After adjusting for all variables, participants who lived with diabetes for 5 years or more were less likely to experience a meaningful decrease compared to participants who had lived with diabetes for less than 5 years (odds ratio [OR] for 5–9 years = 0.18, CI = 0.05, 0.63; OR for 10–19 years = 0.20, CI = 0.06, 0.63; OR for ≥20 years = 0.22, CI = 0.06, 0.76). Additionally, participants who had a significant increase in diabetes self-efficacy score between baseline and three months were more likely to experience a meaningful change compared to participants without this characteristic (Table 2).

## Discussion

In this paper, we sought to conduct a secondary analysis of data from the S4C intervention to identify sociodemographic and disease-related characteristics that influenced how Hispanic/Latinx individuals with T2D responded to S4C. Our

**Table 1. Associations between characteristics of Stories for Change Diabetes participants and meaningful decrease in hemoglobin A1c (HbA1c) from baseline to 3 months (N = 227).[a]**

| | Meaningful decrease in HbA1c from baseline to 3 months | | |
|---|---|---|---|
| | **No (N = 103)** | **Yes (N = 124)** | **P-value** |
| **Sex**, n (%) | | | 0.07[b] |
| Male | 38 (36.9%) | 32 (25.8%) | |
| Female | 65 (63.1%) | 92 (74.2%) | |
| **Age,** (years), n (%) | | | 0.04[b] |
| 18-40 | 9 (9.1%) | 5 (4.0%) | |
| >40-55 | 37 (37.4%) | 66 (53.2%) | |
| >55-70 | 53 (53.5%) | 53 (42.7%) | |
| **Years living with diabetes,** n (%) | | | 0.08[b] |
| <5 | 15 (14.6%) | 34 (27.4%) | |
| 5-9 | 25 (24.3%) | 19 (15.3%) | |
| 10-19 | 40 (38.8%) | 46 (37.1%) | |
| 20+ | 23 (22.3%) | 25 (20.2%) | |
| **Education,** n (%) | | | 0.88[b] |
| High school or more[d] | 28 (27.2%) | 34 (28.1%) | |
| Less than high school | 75 (72.8%) | 87 (71.9%) | |
| **Biometric Data,** Mean (SD) | | | |
| Baseline: HbA1c | 8.9 (1.75) | 9.4 (1.68) | 0.01[c] |
| Baseline: Systolic BP | 126.0 (14.07) | 127.0 (18.34) | 0.93[c] |
| Baseline: Diastolic BP | 76.6 (8.36) | 76.3 (9.55) | 0.59[c] |
| Baseline: BMI | 32.1 (7.27) | 32.8 (7.82) | 0.49[c] |
| Baseline: LDL cholesterol | 95.2 (38.52) | 101.9 (41.34) | 0.21[c] |
| **Social Cognitive Theory Constructs,** Mean (SD) | | | |
| Diabetes Self-efficacy (baseline) | 7.8 (1.48) | 7.1 (1.83) | <.01[c] |
| Diabetes Self-efficacy (3 month) | 8.1 (1.59) | 8.3 (1.39) | 0.74[c] |
| Diabetes Self-efficacy (Difference) | 0.3 (1.51) | 1.1 (1.63) | <.01[c] |
| **Diabetes Social Support** | 2.4 (1.42) | 2.2 (1.45) | 0.35[c] |
| **Culture-Centric Health Promotion Model Constructs,** Mean (SD) | | | |
| Story Identification | 4.9 (0.38) | 4.8 (0.47) | 0.60[c] |
| Story Transportation | 4.9 (0.39) | 4.9 (0.29) | 0.58[c] |
| **Health Literacy,** n (%) | | | 0.65[b] |
| High (score 15–18) | 66 (64.1%) | 83 (66.9%) | |
| Low (≤14) | 37 (35.9%) | 41 (33.1%) | |

[a]Meaningful decrease in HbA1c was defined as a greater than 0.5% decrease from baseline in HbA1c value at 3 month follow up.

[b]Chi-Square p-value

[c]Kruskal-Wallis p-value

[d]The high school or more category included participants who earned a high school diploma, GED, some college or technical school, college or graduate degree.

results suggest that, among S4C intervention participants, those with a diabetes duration of <5 years and/or a significant increase in diabetes self-efficacy between baseline and 3-month follow-up (compared to participants without those characteristics) were more likely to experience a meaningful (>0.5%) decrease in HbA1c at three months. This information is important because it can help us direct this highly scalable intervention to those most likely to derive benefit.

**Table 2. Unadjusted and adjusted odds ratios for associations between Stories for Change Diabetes participant characteristics and meaningful change in hemoglobin A1c (HbA1c) from baseline to 3 months.**[a]

| Variable | Unadjusted | | Adjusted | |
|---|---|---|---|---|
| | Odds Ratio (95% CI) | P-Value | Odds Ratio (95% CI) [b] | P-Value |
| **Sex** | | | | |
| Male | Ref | | Ref | |
| Female | 1.68 (0.95, 2.96) | 0.07 | 1.00 (0.46, 2.20) | 0.99 |
| **Age (years)** | | **0.04** | | 0.13 |
| 18-40 | Ref | | Ref | |
| >40-55 | **3.21 (1.00, 10.29)** | **0.0497** | 5.09 (0.72, 36.05) | 0.10 |
| >55-70 | 1.80 (0.57, 5.73) | 0.32 | 2.82 (0.39, 20.68) | 0.31 |
| **Years living with diabetes** | | 0.09 | | **0.03** |
| <5 | Ref | | Ref | |
| 5-9 | **0.34 (0.14, 0.79)** | **0.01** | **0.18 (0.05, 0.63)** | **0.01** |
| 10-19 | 0.51 (0.24, 1.06) | 0.07 | **0.20 (0.06, 0.63)** | **0.01** |
| ≥20 | 0.48 (0.21, 1.10) | 0.08 | **0.22 (0.06, 0.76)** | **0.02** |
| **Educational Attainment** | | | | |
| High school or more[d] | Ref | | Ref | |
| Less than high school | 0.96 (0.53, 1.72) | 0.88 | 1.48 (0.64, 3.44) | 0.36 |
| **Biometric Data** | | | | |
| Baseline HbA1c | **1.17 (1.00, 1.38)** | **0.047** | 1.26 (1.00, 1.61) | 0.055 |
| Baseline Systolic blood pressure | 1.00 (0.99, 1.02) | 0.66 | 1.03 (1.00, 1.07) | 0.07 |
| Baseline Diastolic blood pressure | 1.00 (0.97, 1.03) | 0.78 | 0.95 (0.90, 1.01) | 0.11 |
| Baseline Body Mass Index | 1.01 (0.98, 1.05) | 0.47 | 1.05 (1.00, 1.11) | 0.06 |
| Baseline LDL Cholesterol | 1.00 (1.00, 1.01) | 0.22 | 1.00 (1.00, 1.01) | 0.35 |
| **Social Cognitive Theory Constructs** | | | | |
| Change in Diabetes Self-efficacy | **1.38 (1.13, 1.69)** | **0.001** | **1.48 (1.17, 1.87)** | **0.001** |
| **Diabetes Social Support** | 0.91 (0.76, 1.09) | 0.29 | 0.96 (0.74, 1.23) | 0.72 |
| **Culture-Centric Health Promotion Model Constructs** | | | | |
| Story identification | 0.78 (0.42, 1.46) | 0.44 | 0.68 (0.28, 1.62) | 0.38 |
| Story transportation | 1.42 (0.65, 3.12) | 0.38 | 1.15 (0.38, 3.44) | 0.81 |
| **Health Literacy** | | | | |
| High (score 15–18) | Ref | | Ref | |
| Low (≤14) | 0.88 (0.51, 1.53) | 0.65 | 0.52 (0.24, 1.13) | 0.10 |

[a]Meaningful decrease in HbA1c was defined as a greater than 0.5% decrease from baseline in HbA1c value at 3 month follow up.

[b]Adjusted for all variables in Table 1: sex, age, years living with diabetes, education, biometric data (HbA1c, blood pressure, BMI, LDL cholesterol), diabetes self-efficacy, story identification and transportation, and health literacy.

[c]Bolded results represent significant associations (p=<0.05).

[d]The high school or more category included participants who earned a high school diploma, GED, some college or technical school, college or graduate degree.

It was unsurprising that participants with a diabetes duration of <5 years were more likely to experience a meaningful decrease in HbA1c. This result could be explained by several factors. First, we know that changing and maintaining new nutrition and physical activity habits after T2D diagnosis is challenging [34]. However, it is less challenging early in the course of the disease than when present for a long time because receiving a diabetes diagnosis may increase patients' awareness of the need to make lifestyle changes [35,36]. Furthermore, patients' diagnosed with T2D motivation to change

behavior may decline over time [37]. Second, while we know that diabetes self-management education (DSME) can successfully support Hispanic/Latinx patients in lowering their HbA1c [38], recently diagnosed patients are more likely to be referred to DSME compared to those with longer T2D duration. Thus, participation in DSME outside our study (something we did not document) may have influenced the recently diagnosed patients. Third, it is also possible that those with T2D for ≥5 years had more complex treatment regimens meaning a) it would have been harder for them to lower their HbA1c with health behavior change alone and/or b) they had even higher HbA1c targets due to clinical complexity. Therefore, while more research is needed, this finding may indicate that S4C could be a good addition to T2D diagnosis educational offerings but likely needs to be adapted for those with longer disease duration from the Hispanic/Latinx population. Additionally, tailoring interventions based on an individual's disease duration might optimize outcomes.

It was also unsurprising that participants with a significant increase in diabetes self-efficacy were more likely to experience a meaningful decrease in HbA1c. Research has demonstrated that self-efficacy can play a mediating role in the relationship between diabetes education and self-care. This means diabetes education can lead to a higher likelihood that a patient engages in self-care activities and this relationship is at least partially explained by an increase in self-efficacy [39]. Thus, the S4C participants with a significant increase in diabetes self-efficacy may have responded more positively to the intervention because they were confident in their ability to learn and then complete diabetes-related self-care activities. Future interventions could focus on mechanisms to boost self-efficacy, particularly in populations that were newly diagnosed or those with lower initial self-efficacy to maximize the potential benefits of DST interventions.

There were two unexpected findings. First, in contrast to our hypothesis, the S4C intervention did not have a greater impact on individuals who had less formal education. This may have occurred because most participants had less than a high school education, meaning the comparison group of individuals with more formal education may not have been large enough. Second, based on our theoretical model we anticipated that story identification and story transportation would be significantly higher for participants who experienced a meaningful decrease in HbA1c compared to those who did not. Yet this did not occur. We may have gotten this result because both groups reported high story identification and transportation or the NQAT may have ceiling effects. Going forward, more research is needed to further investigate how DST interventions impact individuals with more than a high school education as well as story identification and transportation (e.g., by asking the NQAT after each digital story rather than a group of digital stories). The specific mechanisms of the DST process and how digital stories might influence longitudinal behavior change could be evaluated in a future clinical trial.

Our findings are similar to the work of others. Houston et al. conducted a randomized control trial that compared the impact of a video of patient stories about hypertension to a video covering unrelated health topics among African American individuals with hypertension. The authors found that patients in the intervention group with uncontrolled hypertension at baseline decreased their blood pressure more than their peers in the control group. In contrast, patients who entered the study with controlled hypertension did not significantly differ over time. While blood pressure subsequently increased among both groups, the relative advantage among intervention participants remained through the end of the study [26]. Similarly, in the pilot study that led to the parent study that this analysis is based on, Wieland et al. found that Hispanic/Latinx patients who had a baseline HbA1c of >7% experienced a larger decrease in HbA1c compared to Hispanic/Latinx patients who had a baseline HbA1c of ≤7% [16]. The present study contributes to this line of inquiry by adding more evidence explaining which disease and/or demographic characteristics may impact how Hispanic/Latinx individuals respond to a DST intervention. With this information, we can further tailor DST interventions and thereby enhance effectiveness.

## Limitations

One limitation of this study was that we did not control for T2D comorbidities, complications, treatment plans, or concurrent DSME. Additionally, most of our participants identified as female (69%). More research is needed to investigate the impact of S4C on Hispanic/Latinx adults who identify as male. Finally, this study is limited by the relatively brief follow-up period of three months for T2D-related change. Future studies should examine the longer-term impact of DST

on HbA1c and other health outcomes. Despite these limitations, our research fills an important gap in knowledge around which sociodemographic and disease characteristics influence viewers response to S4C. These findings will guide future research to assess the mechanisms by which these factors influence how viewers respond to DST interventions.

## Conclusions

Because culturally relevant digital stories have been demonstrated to cause positive behavior change [5], further tailoring these interventions for specific patient populations would be expected to provide a more effective approach compared to more general interventions. The findings from this study will have the potential to inform the development of effective interventions to reduce disparities in T2D burden among Hispanic/Latinx populations. Furthermore, tailored digital stories can be seamlessly integrated into a range of T2D educational resources, as they are a low-cost, portable, and easy-to-deliver intervention that does not create extra work for healthcare providers. Thus, the results of this research will allow us to efficiently direct this highly scalable intervention to patients and communities most likely to benefit.

## Author contributions

**Conceptualization:** Abby M. Lohr, Jennifer St. Sauver, Joan M. Griffin, Rozalina G. McCoy, John M. Ruiz, Irene G. Sia, Mark L. Wieland.

**Data curation:** Jamie O'Byrne.

**Formal analysis:** Abby M. Lohr, Jennifer St. Sauver, Jamie O'Byrne.

**Funding acquisition:** Abby M. Lohr, Joan M. Griffin, Rozalina G. McCoy, John M. Ruiz, Irene G. Sia, Mark L. Wieland.

**Project administration:** Abby M. Lohr.

**Supervision:** Sunny W Kim, Jennifer St. Sauver, Joan M. Griffin, Rozalina G. McCoy, John M. Ruiz, Irene G. Sia, Mark L. Wieland.

**Writing – original draft:** Abby M. Lohr, Sunny W Kim.

**Writing – review & editing:** Joan M. Griffin, Rozalina G. McCoy, John M. Ruiz, Irene G. Sia, Mark L. Wieland.

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
