## [Decision Letter · Decision Letter 0]

12 Mar 2025

Dear Dr. Lohr,

We look forward to receiving your revised manuscript.

Kind regards,

Guoying Wang, MD, PhD

Academic Editor

PLOS ONE

Additional Editor Comments (if provided):

Reviewers' comments:

Reviewer's Responses to Questions

**Comments to the Author**

1. Is the manuscript technically sound, and do the data support the conclusions?

Reviewer #1: Partly

Reviewer #2: Yes

2. Has the statistical analysis been performed appropriately and rigorously?

Reviewer #1: Yes

Reviewer #2: Yes

3. Have the authors made all data underlying the findings in their manuscript fully available?

Reviewer #1: Yes

Reviewer #2: Yes

4. Is the manuscript presented in an intelligible fashion and written in standard English?

Reviewer #1: Yes

Reviewer #2: Yes

Reviewer #1: The authors evaluate sociodemographic and disease characteristics of a Hispanic/Latinx population with DM which may influence the effect of digital storytelling for change of diabetes upon the HbA1c levels.

They found that in the studied cohort people with a disease duration of less than 5 years and those who had an improvement of diabetes self-efficacy between baseline and 3 months were more likely to experience a

meaningful decrease in hemoglobin A1c at three months

The paper is well written. However, I do not agree with the authors' conclusion that "... our research fills an important gap in knowledge around why some individuals respond with behavior change and healthful behaviors after viewing a digital story and others do not." In fact, the study only showed that people with the above-mentioned characteristics respond better, but it did not evaluated why they respond better - although some hypothesis were discussed. Actually, this should be the aim of a new research - to investigate which factors (phychological and external) influence these responses, since many people who had not the above-described characteristics still had an improvement in Hba1c.

"The study findings will contribute (not allow) to digital storytelling interventions to be more effectively tailored to those most likely to benefit". They also indicate the population with better responses to SC in which future research should be undertaken in order to find out which characteristics of the S4C interventions induced positive effects, allowing to improve these storytelling narratives for the benefit of a larger population (which is the larger goal of this type of intervention).

Reviewer #2: This is a secondary analysis to examine the associations of the sociodemographic and disease related factors with participants’ responses to the Stories for Change Diabetes intervention in the 227 intervention participants. The study found that participants with diabetes duration <5 years and/or whose diabetes self-efficacy improved between baseline and 3-month follow-up were more likely to experience a meaningful decrease in hemoglobin A1c at three months. This study is interesting.

Comments

1. There is a lack of Ethics Statement in the text.

2. Since the S4C project included 450 participants, it is unclear why this study only included 227 participants. It is also unclear if the selection led to bias.

3. It will be helpful to add the sample size and number of cases who experience a meaningful decrease in HbA1c for each group in Table 2.

4. Given a small number in the 18-40 years age group (the reference group), it is unclear if the regression model is a good fit.

5. The methods for measuring blood pressure, BMI, lipids, and HbA1C were not provided.

**Do you want your identity to be public for this peer review?** For information about this choice, including consent withdrawal, please see our Privacy Policy

Reviewer #1: No

Reviewer #2: No

---

## [Author Response · Author response to Decision Letter 1]

21 Mar 2025

Response to Reviewers

Reviewer #1: The authors evaluate sociodemographic and disease characteristics of a Hispanic/Latinx population with DM which may influence the effect of digital storytelling for change of diabetes upon the HbA1c levels.

They found that in the studied cohort people with a disease duration of less than 5 years and those who had an improvement of diabetes self-efficacy between baseline and 3 months were more likely to experience a meaningful decrease in hemoglobin A1c at three months.

The paper is well written. However, I do not agree with the authors' conclusion that "... our research fills an important gap in knowledge around why some individuals respond with behavior change and healthful behaviors after viewing a digital story and others do not." In fact, the study only showed that people with the above-mentioned characteristics respond better, but it did not evaluated why they respond better - although some hypothesis were discussed. Actually, this should be the aim of a new research - to investigate which factors (phychological and external) influence these responses, since many people who had not the above-described characteristics still had an improvement in Hba1c.

Thank you for bringing this to our attention. We agree, the original statement overreaches. We’ve changed this sentence to read: “Despite these limitations, our research fills an important gap in knowledge around which sociodemographic and disease characteristics influence viewers response to S4C. These findings will guide future research to assess the mechanisms by which these factors influence how viewers respond to DST interventions.”

"The study findings will contribute (not allow) to digital storytelling interventions to be more effectively tailored to those most likely to benefit". They also indicate the population with better responses to SC in which future research should be undertaken in order to find out which characteristics of the S4C interventions induced positive effects, allowing to improve these storytelling narratives for the benefit of a larger population (which is the larger goal of this type of intervention).

Thank you for this suggestion. For flow, we changed this sentence to read: “These findings will provide insight into how digital storytelling interventions can be effectively tailored to Hispanic/Latinx individuals most likely to benefit.”

Reviewer #2: This is a secondary analysis to examine the associations of the sociodemographic and disease related factors with participants’ responses to the Stories for Change Diabetes intervention in the 227 intervention participants. The study found that participants with diabetes duration <5 years and/or whose diabetes self-efficacy improved between baseline and 3-month follow-up were more likely to experience a meaningful decrease in hemoglobin A1c at three months. This study is interesting.

Comments

1. There is a lack of Ethics Statement in the text.

Thank you for this comment in line 127 we stated: “Both the parent and current studies were approved by the Mayo Clinic Institutional Review Board (18-002998 and 22-008310).”

2. Since the S4C project included 450 participants, it is unclear why this study only included 227 participants. It is also unclear if the selection led to bias.

Thank you for this feedback. In lines 107-111, we explained that we only used the S4C randomized control trial intervention participant data in the described secondary analysis because this was the only group who received the intervention. We added our sample size in line 108 to clarify. Because these participants were randomized, we were able to avoid selection bias.

“In this investigation, we use the S4C randomized control trial data to conduct a secondary analysis of the participants who received the intervention (n=227) to identify sociodemographic and disease status characteristics that influenced how Hispanic/Latinx individuals with T2D responded to S4C. Control group participants, who did not view the intervention, were excluded from this analysis.”

3. It will be helpful to add the sample size and number of cases who experience a meaningful decrease in HbA1c for each group in Table 2.

Thank you for this suggestion. Table 2 is the summarization of modelled output from the tests of association between various potential explanatory variables and whether a subject experienced a meaningful decrease in HbA1c. While we feel it wouldn’t be appropriate to include the actual counts of cases who experience a meaningful decrease in HbA1c for each group alongside modelled output in Table 2, that information can be found in Table 1.

4. Given a small number in the 18-40 years age group (the reference group), it is unclear if the regression model is a good fit.

Thank you for this question. While we agree that it’s unfortunate from a modelling perspective to have so few subjects in the 18-40 age group, this group was chosen as the reference to provide a natural comparison group for the groups of older subjects, as the group most likely to generate ORs for the other 2 groups that are both greater than 1. Further, neither the unadjusted nor adjusted models returned unbounded estimates for the confidence intervals for the ORs which would normally prompt removal of a troublesome subgroup. Addtionally, Hosmer-Lemeshow tests for goodness-of-fit of the multivariable model showed no reason to suspect the model was a poor fit to the data (p-value=0.7448).

5. The methods for measuring blood pressure, BMI, lipids, and HbA1C were not provided.

Thank you for this feedback. We added more detail on how we measured blood, pressure, BMI, and lipids.

We collected additional biometric data including blood pressure, body mass index (BMI), and cholesterol. Study personnel recorded seated blood pressure on participants' right arms after they had rested quietly for five minutes. Blood pressure was measured three times, and the average of the second and third readings was used for analysis. Weight was measured to the nearest 0.1 kg using a portable scale, and height was measured to the nearest 0.1 cm at baseline using a stadiometer, with participants removing their shoes for both measurements. Body mass index (BMI) was calculated as weight (kg) divided by height squared (m²). Low-density lipoprotein (LDL) cholesterol was measured using the same blood sample that was utilized to determine HbA1c levels. LDL cholesterol was calculated for each participant based on these values: LDL-cholesterol = total cholesterol - HDL cholesterol - (triglycerides/5). All measurements were collected for the study (and hence free to the patient) rather than for clinical care.

In lines 145-8, we described how we measured HbA1C: “The primary outcome was glycemic management as assessed by HbA1c measured from blood samples obtained and analyzed by the laboratories at the participating sites. A meaningful decrease in HbA1c was defined as a greater than 0.5% reduction from baseline to 3-month follow up.”

We made one slight, additional change to the manuscript which was to add in the results for the diabetes social support scale into both Table 1 and 2 as well as a short paragraph describing this variable in the methods section. The reason for this is that the model being shown in the multivariable/adjusted model output was output from a model including diabetes social support. Given that other parameters would have been changed if this variable was excluded, we assumed it was removed in error. We’ve added it back in for completeness.

---

## [Decision Letter · Decision Letter 1]

29 Apr 2025

Exploring factors impacting Hispanic/Latinx individuals’ response to a type 2 diabetes digital storytelling intervention

PONE-D-25-02358R1

Dear Dr. Lohr,

We’re pleased to inform you that your manuscript has been judged scientifically suitable for publication and will be formally accepted for publication once it meets all outstanding technical requirements.

Kind regards,

Guoying Wang, MD, PhD

Academic Editor

PLOS ONE

Additional Editor Comments (optional):

Reviewers' comments:

Reviewer's Responses to Questions

**Comments to the Author**

Reviewer #1: All comments have been addressed

Reviewer #2: All comments have been addressed

2. Is the manuscript technically sound, and do the data support the conclusions?

Reviewer #1: Yes

Reviewer #2: Yes

3. Has the statistical analysis been performed appropriately and rigorously?

Reviewer #1: Yes

Reviewer #2: Yes

4. Have the authors made all data underlying the findings in their manuscript fully available?

Reviewer #1: Yes

Reviewer #2: Yes

5. Is the manuscript presented in an intelligible fashion and written in standard English?

Reviewer #1: Yes

Reviewer #2: Yes

Reviewer #1: (No Response)

Reviewer #2: Thank authors response to my comments. My comments has been well addressed. I don't have any other comments.

**Do you want your identity to be public for this peer review?** For information about this choice, including consent withdrawal, please see our Privacy Policy

Reviewer #1: No

Reviewer #2: No

---

## [Editor Report · Acceptance letter]

PONE-D-25-02358R1

PLOS ONE

Dear Dr. Lohr,

I'm pleased to inform you that your manuscript has been deemed suitable for publication in PLOS ONE. Congratulations! Your manuscript is now being handed over to our production team.

Kind regards,

on behalf of

Dr. Guoying Wang

Academic Editor

PLOS ONE